# Determinants of agricultural employment during the COVID-19 pandemic: A spatial analysis of Brazilian municipalities

Patrícia Batistella[1], Luan Marca[1]*, Fernanda Castilhos França de Vasconcellos[2], Eduardo Rodrigues Sanguinet[3], Augusto Mussi Alvim[1]*, Adelar Fochezatto[1]

1 School of Business, Pontifical Catholic University of Rio Grande do Sul (PUCRS), Porto Alegre, RS, Brazil, 2 Program in Rural Development (PGDR), Federal University of Rio Grande do Sul (UFRGS), Porto Alegre, RS, Brazil, 3 Institute of Agricultural Economics, Universidad Austral de Chile, Valdivia, Chile

* augusto.alvim@pucrs.br (AMA); luan.marca@edu.pucrs.br (LM)

**Data Availability Statement:** All data are in the manuscript and supporting information files".

## Abstract

The impact of COVID-19 has extended beyond the health toll it has taken on populations. The global economy has experienced significant downturns, with unemployment rates reaching unprecedented highs for this century. Nonetheless, the agricultural sector has been uniquely affected by the pandemic, particularly given its crucial role in food supply. This study sought to examine the effects of the COVID-19 pandemic on agricultural employment in Brazilian municipalities. To achieve this, we employed a dataset spanning the period before and during the pandemic (2018–2021). Spatial econometrics was utilized to identify potential spatial factors linked to this occurrence. Notably, the findings reveal that the transfer policies implemented by the Brazilian Federal Government have had a positive impact on agricultural employment in Brazilian municipalities, challenging the common assumption that such measures might discourage job-seeking efforts. Moreover, the results suggest that the agricultural sector has managed to absorb individuals displaced from other sectors of the economy, demonstrating remarkable resilience in the face of economic challenges.

## Introduction

During the COVID-19 pandemic, unemployment levels increased across various regions and economic sectors in Brazil. Between the last quarter of 2019, before the pandemic was declared, and the first quarter of 2021, Brazil's unemployment rate rose from 11.1% to 14.9%, the highest recorded since 2012 [1]. The main drivers of this increase were the restrictive measures adopted by regional and national governments aimed at reducing the movement of people, which had a heterogeneous impact on employment levels across sectors [2]. Although sectors deemed essential, such as agriculture, faced fewer direct restrictions, they also experienced employment variations [3, 4].

Despite the economic importance of the agricultural sector in the Brazilian economy, representing about 23.8% of the GDP and 26.8% of the workforce [5], there was a decline in employment, particularly among less qualified workers [6]. Although mobility restrictions did not directly affect agricultural employment, they impacted the supply chain, especially on the

**Funding:** Agencia Nacional de Investigacion y Desarrollo. The funders had no role in study design, data collection and analysis, decision to publish, or preparation of the manuscript.

**Competing interests:** The authors have declared that no competing interests exist.

demand side, causing disruptions in production chains and affecting the distribution of goods [7, 8]. Moreover, the lockdowns and mobility restrictions implemented to contain the virus's spread had indirect effects on agricultural work due to its essential and seasonal nature [9]. Conversely, the demand for fresh food and horticultural products, along with increased exports of various commodities, created seasonal work opportunities for those who lost jobs in other sectors [10].

The objective of this article is to estimate the determinants of regional agricultural employment variation in Brazil between 2018 and 2021. The contribution of this study is threefold. First, it assesses the impact of socioeconomic determinants, public policies, COVID-19-related events, and extreme weather events. This provides regional empirical evidence on the main factors associated with employment variation during the period of economic restrictions, allowing for a detailed analysis of how the labor market responded to multiple dimensions. Secondly, the study incorporates spillover effects at the municipal level, identifying both the direct and indirect impacts of these factors during the pandemic, making it possible to distinguish how agricultural employment spatially disperses in response to different variables. Finally, the analysis allows for regional differentiation of how various factors—from local economic characteristics to climatic events and the effects of the pandemic—heterogeneously affect agricultural employment in municipalities.

The analysis of the determinants of employment variation in the agricultural sector needs to be multifactorial. The agricultural supply chain is influenced by both market aspects [11, 12], such as productivity levels and export demand [13, 14], and climatic characteristics, which affect its ability to respond to market dynamics [15]. Furthermore, employment in the sector is impacted by external factors, such as the spread of COVID-19. In this context, the research aims to differentiate the relative contribution of each of these dimensions to regional employment, considering that the agricultural sector is strongly interconnected with the rest of the economy due to its essential nature.

The evidence on the determinants of municipal-level agricultural employment variation in Brazil does not account for local spillover effects. Agriculture, due to its economic importance, significantly influences employment variations, especially in specialized areas where the labor market is directly linked to the sector and generates effects in neighboring regions [16]. Spatially localized extreme weather events affect both municipal agricultural production and areas interconnected through supply chains [17, 18]. Additionally, socioeconomic factors, such as income transfers, and external disruptions, such as the pandemic, impact the demand for agricultural labor and its availability [19, 20]. Given this interaction of factors, a spatial analysis is necessary to fill this gap in the literature, as it allows us to understand how variables affect different regions and identify spatial patterns of variation in agricultural employment.

The study specifies a spatial panel model that captures the spatial interdependencies between municipalities across all incorporated dimensions [21]. Additionally, the model considers both spatial dependence and heterogeneity among the units of analysis throughout the studied period. The temporal control covers two periods: 2018–2019, before the pandemic, and 2019–2021, during the pandemic. In this context, the article highlights the resilience of the agricultural sector during crises, identifying how multiple factors affect the vulnerability of local agricultural employment in a country with pronounced spatial inequalities, such as Brazil.

This article is divided into five sections, including this introduction. The second section presents a literature review on the relationship between the COVID-19 pandemic and agricultural employment, as well as discusses the role of spatial models in addressing the study's problem. The third section details the methodological steps of the research, while the fourth presents the main results. Finally, the last section provides conclusions.

## Current context

This section examines the economic impacts of COVID-19 on agricultural employment, addressing the influence of additional variables on agricultural employment in Brazil. Furthermore, the section also discusses results from similar studies that also use spatial analysis. Through this examination, we aim to identify the variables most likely to significantly affect agricultural employment, as indicated by current literature. Finally, we present a profile of agricultural production in the different regions of Brazil.

**Impact of the COVID-19 pandemic on agricultural employment.** The agricultural supply chain experienced disruptions due to mobility restrictions, lockdowns, and border closures. These disruptions particularly affected the hiring of seasonal workers, who play a crucial role in harvesting and other labor-intensive agricultural activities [7, 8]. According to Gray [9], the travel restrictions imposed significantly reduced the availability of seasonal labor, impacting the harvesting of fresh produce and horticultural products, especially in emerging economies and countries heavily dependent on agricultural exports, such as Brazil and the United States.

The agricultural sector was less affected by the pandemic compared to the industrial and service sectors, partly due to being considered essential, which allowed many activities to continue even during periods of severe restrictions [22, 23]. According to Khamis *et al.* [24], the income reduction for small family farmers was significant but less severe than for urban entrepreneurs, who faced challenges with business continuity and cash flow. The adaptability of family farmers, such as diversifying production to meet local demand, helped mitigate the effects of the crisis. Furthermore, the labor shortage in the field caused by mobility restrictions and the inability to hire foreign seasonal workers resulted in a temporary migration of workers from the service and industrial sectors to agriculture [25].

This migration of workers to agriculture may have been more pronounced in developing countries. Deb [26] indicates that with factory closures and disruptions in supply chains in urban areas in India, many workers, especially informal and self-employed, lost their jobs and returned to their hometowns, where agriculture continued to provide some form of livelihood. This movement generated an increased demand for jobs in the rural sector as job opportunities in urban areas drastically diminished.

Despite the adversities, one of the contradictory effects of the pandemic was the increased demand for certain fresh foods and horticultural products. The closure of restaurants and the shift to home consumption drove this demand in many countries [27]. Additionally, agricultural products aimed at export, such as grains and meat, continued to play a central role in global food supply chains. Despite mobility restrictions and logistical difficulties, the demand for agricultural commodities remained robust, especially in Asian markets [28–30]. Specific agricultural sectors, such as horticulture and seasonal crop harvesting, began to demand more labor to adapt to the new market dynamics [31].

Regarding Brazil, the results of agriculture during the pandemic were positive, highlighted by a record increase in exports, especially soybeans, sugarcane derivatives, meats, cotton, vegetables, fats, and vegetable oils [28]. The devaluation of the Brazilian real against the dollar, coinciding with rising commodity prices in the international market, benefited exporters [32]. In some regions of Brazil, agricultural activity recorded an increase in the number of employed individuals, primarily among those with higher education, while less qualified workers suffered more from the impacts [6], reinforcing trends observed in developed countries, where the pandemic hit lower-educated workers harder [3, 4, 33].

Although some factors negatively affected Brazilian agriculture, such as extreme weather events, rising fuel prices, and difficulties in importing production inputs (pesticides and

fertilizers), there was an increase in production during this period, even amidst a public health crisis [34].

Based on the evidence from the literature regarding the impact of the pandemic on agricultural employment, the following research hypothesis is proposed:

$H_1$: The pandemic, in terms of the number of COVID-19 cases, positively impacted agricultural employment in Brazil.

**Application of spatial econometric models.**   Studies investigating the effects of the pandemic on agricultural variables using spatial regression methods are rare, especially in the Brazilian context. International literature provides some examples that highlight the importance of considering spatial aspects in analyzing these impacts. Yao *et al.* [35] conducted a pioneering study by using geographically weighted regression methods (GWR and MGWR) to examine the effects of the pandemic, economic recession, and climate change on food crop production and global food security, demonstrating that the direct effects of COVID-19 on agricultural production were small and localized. Similarly, Wicaksana and Fitrady [16] examined the spatial effects of agricultural policies on agricultural prices during the pandemic, revealing an intensification of spatial dependence during the pandemic period.

Conversely, Yeboah *et al.* [36] revealed that the pandemic increased food insecurity by 33.5% in 2020 in the United States, with poverty emerging as the main determinant. The authors highlighted that although government aid alleviated this impact, spatial analysis showed that residents of the Western U.S. were the most affected, regardless of race. In Brazil, there is a significant gap in the literature exploring the spatial effects of the pandemic on agricultural variables. Based on this gap, the following hypothesis is proposed:

$H_2$: The effects of the COVID-19 pandemic on agricultural employment present spatial spillover effects, both at the macro and regional levels.

## Determinants of agricultural employment

Agricultural employment is influenced by a variety of economic, social, and environmental factors, which can both drive and reduce the demand for labor in rural areas. In this regard, rural credit access, transport infrastructure, and the capacity for technological innovation stand out [11, 12, 37, 38]. Agricultural modernization, driven by the adoption of new technologies, can both create new employment opportunities and reduce the need for labor in certain activities [13, 39]. Mechanization and automation, for instance, are gradually replacing manual labor in rural areas, potentially reducing the demand for workers. However, they also create new needs for professional qualifications and technology management.

Brazilian agriculture, especially in regions dependent on crops like soybeans and corn, is highly susceptible to climate variability [18, 15, 40]. Phenomena such as prolonged droughts, severe dry spells, floods, and frosts affect production, leading to reduced labor demand during critical periods [17]. These events can disrupt the production cycle, affecting not only the creation of seasonal jobs but also the income and food security of families [41, 42]. Studies such as Easterling *et al.* [43] and Harvey *et al.* [44] show that the frequency and intensity of extreme weather events have increased in recent decades, and with this, the risk to agricultural employment security.

In addition to climate factors, public income transfer policies implemented in Brazil—such as *Bolsa Família*, a conditional cash transfer program targeting families in poverty and extreme poverty; the *Continuous Cash Benefit (BPC)*, which ensures a minimum monthly income for

elderly individuals aged 65 or older and for people with disabilities; and *Emergency Aid*, a temporary program created during the COVID-19 pandemic to support vulnerable populations and mitigate its economic impacts—played a crucial role in sustaining rural families' income during the pandemic. Among these programs, *Bolsa Família* was particularly effective in reducing extreme poverty in rural areas, serving as a vital safety net during the crisis. Likewise, *Emergency Aid* provided financial relief to families experiencing reduced income due to mobility restrictions and declining demand for certain agricultural products. Without these initiatives, the socioeconomic vulnerability of rural areas would have significantly worsened [19, 20].

In this context, the following research hypotheses are proposed:

$H_3$: Extreme weather events had a significant impact on the reduction of agricultural employment during the period.

$H_4$: The implementation of public income transfer policies during the health crisis positively influenced the maintenance of agricultural employment.

**Regional characteristics of the agricultural sector in Brazil.**   Brazil's continental dimensions and diversity of climates and soils present a complex and rich agricultural structure that varies significantly from one region to another. Fig 1 illustrates the country's five major regions and their primary agricultural products. Each of these regions has distinct characteristics that influence both agricultural production and employment [14]. Understanding these differences is crucial for analyzing the impacts of the COVID-19 pandemic on Brazil's agricultural sector.

The North region, comprising the states of the Amazon, has vast expanses of tropical forest. The local agricultural sector, though significant, faces a series of challenges related to balancing economic development with environmental preservation. Agricultural production is primarily focused on grains, tropical fruits, and agro-industrial products such as açaí, cocoa, and Brazil nuts [45]. Livestock farming is also gaining importance, particularly in states like Pará and Rondônia, which show high deforestation rates due to pasture expansion for cattle ranching [46]. Additionally, agricultural labor in this region is relatively limited compared to other regions of Brazil, due to low population density and limited service infrastructure [47].

Agriculture in the Northeast region is characterized by a diversity of crops, with an emphasis on subsistence farming, including cassava, corn, and beans. However, there are also areas dedicated to agribusiness, particularly in soybean and sugarcane production, concentrated in states like Bahia and Alagoas [45]. One of the main challenges of agriculture in this region is climatic variability. The semi-arid region, which covers much of the Northeast, is affected by prolonged droughts and irregular rainfall, making agricultural production a high-risk activity. Employment in the sector has a high proportion of informal and seasonal workers, with significant challenges related to income and working conditions [48, 49].

The Midwest region is one of the pillars of Brazilian agriculture, with a focus on grain production, especially soybeans and corn, in addition to an extensive livestock sector. Technological advancements and the expansion of cultivated areas in the *cerrado* biome—recognized as the savanna with the greatest biodiversity in the world, home to approximately 11,627 species of native plants, of which around 4,400 are endemic—have boosted productivity and competitiveness in the region [50, 51]. Agricultural labor in this area tends to be more qualified due to mechanization and the adoption of advanced technologies [52]. Furthermore, the development of transportation infrastructure and the integration of production chains have also contributed to the sector's growth.

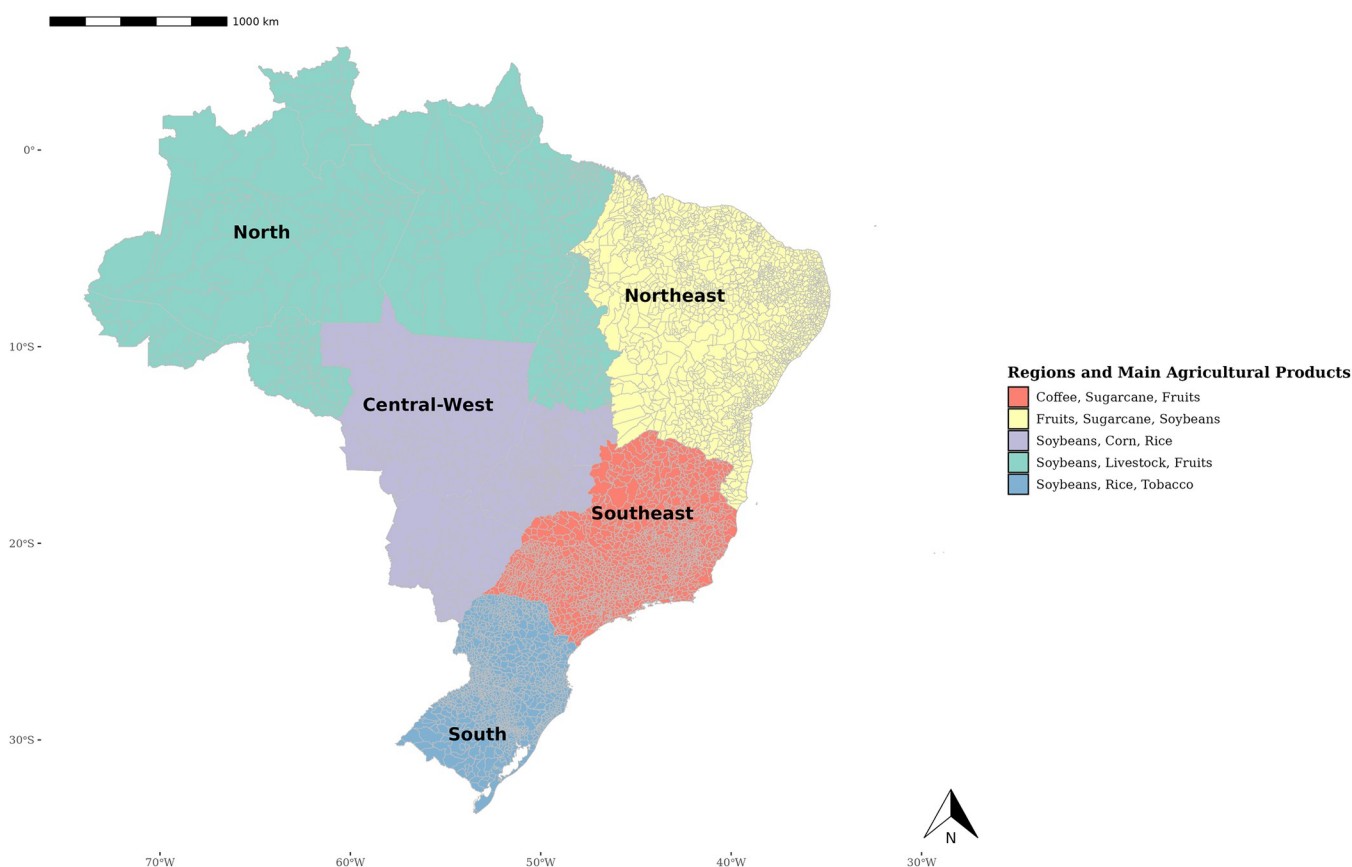

**Fig 1. Brazilian regions.** Source: The map was created using the ggplot2 and geobr packages available for the R software. The geobr package provides georeferenced data of Brazilian municipalities made available by the Brazilian Institute of Geography and Statistics (IBGE). The data plotted on the map were obtained from the Municipal Agricultural Survey (PAM), conducted by IBGE. All data used are publicly available and compatible with the CC BY 4.0 license.

The economically developed Southeast region has a diversified agricultural sector that includes coffee, sugarcane, and vegetable production, along with well-developed livestock farming [45]. The region's agriculture is capital- and technology-intensive, reflecting its proximity to large urban centers and markets [53]. Agricultural labor is predominantly composed of more qualified workers, due to the presence of agro-industrial companies and cooperatives that offer better working conditions and training [54]. Similarly, the South region stands out for its variety of crops, including soybeans, corn, rice, and tobacco, as well as a well-established dairy and beef livestock sector. The temperate climate and fertile soils contribute to high productivity [55]. Working conditions and investments in agricultural technology have been positive factors.

Based on the characteristics of each region, the following hypothesis is proposed:

$H_5$: The effects of the pandemic on agricultural employment are heterogeneous, varying according to the productive characteristics of each region of the country.

Fig 2 presents a visual scheme that facilitates the understanding of the multiple hypotheses incorporated in the study. It highlights the main factors influencing the variation in agricultural employment in Brazil before and during the COVID-19 pandemic. On the time axis, agricultural employment is represented at the initial point ($t = 0$) and during the pandemic ($t = 1$). Between these two points, different dimensions are shown: the health dimension

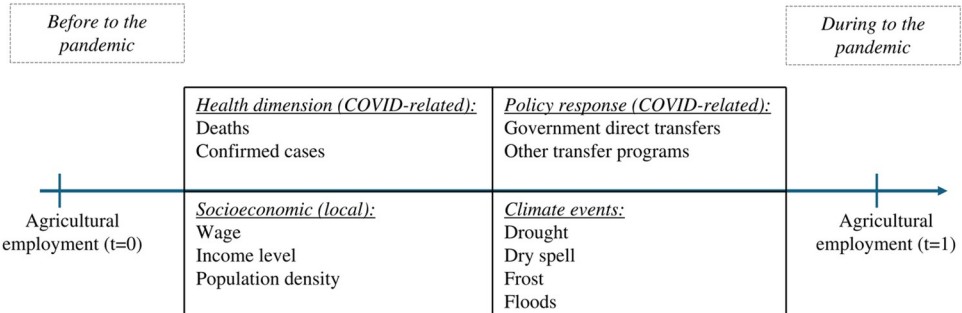

**Fig 2. Main assumptions and factors affecting agricultural employment in Brazil during the COVID-19 pandemic.** Source: prepared by the authors.

related to COVID-19 (confirmed cases and deaths), policy responses (direct transfers and aid programs), local socioeconomic factors (wages, income level, and population density), and climate events (droughts, frosts, and floods). These elements are analyzed in the study to understand how they heterogeneously and spatially influence the dynamics of agricultural employment over time.

## Materials and methods

This section describes the spatial panel data models along with the assessments employed to identify the most stable models. For this, we defined the spatial matrices, Moran's I, fitted spatial models, implemented tests, and ultimately, an overview of the database utilized.

### Spatial weight matrix

The starting point for spatial regression analysis is determining the spatial weight matrix. This matrix explicitly establishes the spatial interdependence among the regions (observations) under analysis [56, 57]. In our study, we posit that the spatial weights matrix is external to the econometric model, allowing it to be articulated through a matrix $W$, in which $W_{i,j}$ represents the matrix derived from the inverse function of the distance between the centroids of the regions, utilizing $1/d_{ij}^2$ for $i$ and $j$ spatial units. The employment of the inverse of the squared distance function ($d_{i,j}$) mirrors the gravitational function, as highlighted by Dall'erba and Gallo [58] and Bivand *et al.* [59].

The decision to use a matrix with these characteristics was driven by two main factors. First, this approach acknowledges that, in practice, spatial interactions often follow a gravitational decay logic, where physical proximity implies greater influence, especially in economic matters like agricultural employment, which is sensitive to local and regional conditions. Second, this choice allows for a more precise capture of spatial externalities, reflecting the impact that public policies, climatic conditions, and other local factors may have on neighboring regions. The use of distance-based matrices has been widely applied in analyses of economic phenomena [60–62].

The neighborhood matrix used in the analysis covers all 5,570 municipalities in Brazil and contains 29,730 non-null connections, representing spatial interactions between the municipalities. On average, each municipality is connected to about 5.34 other municipalities, although two regions show no connections (islands). The construction of the matrix was based on georeferenced data provided by the Brazilian Institute of Geography and Statistics (IBGE) and accessed through the geobr package for the R programming language [63].

## Spatial correlation test: Moran's index

The test used to verify the spatial correlation between the ratio of new agricultural jobs and total employment in Brazilian municipalities is Moran's I. This index aims to estimate the correlation between the values of a variable in a given location and the values in neighboring areas. Moran's I shows whether activities are concentrated or not. A value above zero indicates that the variable y is spatially and positively correlated between neighboring municipalities; a value below zero suggests a negative spatial correlation between neighboring municipalities. Finally, a large (small) absolute value of Moran's I points to a strong (weak) spatial correlation.

## Spatial panel models

This research employs spatial panel data models to account for the spatial correlation effects among the municipalities under study. It examines three spatial models: the spatial autoregressive model (SAR), the spatial error model (SEM), and the spatial Durbin model (SDM), as outlined by Elhorst [21].

The SAR model considers the effects of the spatially lagged dependent variables ($Wy$), particularly on the dependent variable (y). The model posits that spatial effects can be inferred through the correlation between y and Wy across different Brazilian municipalities. The formulation of the SAR model is as follows (Eq 1):

$$y = \rho W y + X\beta + \varepsilon \tag{1}$$

Where p is the spatial autocorrelation coefficient, $\varepsilon$ is the error term ($\varepsilon \sim N(0, \sigma^2)$), and X is the vector of explanatory variables within the model.

Conversely, SEM pertains to a spatially defined model where the errors corresponding to each observation i exhibit correlation with the errors associated with other observations j. This model can be expressed using Eqs 2 and 3:

$$y = X\beta + \mu \tag{2}$$

$$\mu = \lambda W \mu + \varepsilon \tag{3}$$

Where $\lambda$ is the spatial autocorrelation coefficient, and $\mu$ is the error with spatial autocorrelation between different municipalities.

Lastly, the SDM incorporates both lagged dependent variables (Wy) and lagged independent variables (Wx). This spatial model is seen as an unconstrained form of the SAR and SEM and is expressed as follows (Eq 4):

$$y = \rho W y + X\beta + \gamma W X + \varepsilon \tag{4}$$

Where $\gamma$ is the spatial autocorrelation coefficient associated with the independent variables, and the other parameters and variables as defined above.

## Selection of spatial panel models

The selection of the appropriate model to analyze the data requires careful consideration of the presence of unobserved heterogeneity. It is essential to evaluate fixed effects (FE) and random effects (RE) models to determine their suitability to the data structure. This evaluation is conducted using the Hausman test, which checks whether unobserved heterogeneity is correlated with the explanatory variables. As formulated by Baltagi [64], this test expresses the null hypothesis $\beta^{EA} = \beta^{EF}$, testing the orthogonality between individual effects and exogenous

variables. The test statistic W follows a chi-squared distribution, considering the degrees of freedom (k) of the matrix ($\beta^{EA}$–$\beta^{EF}$).

If $H_0$ is not rejected, it suggests that $E(a|X) = 0$, indicating that the random effects ($\beta^{EA}$) are consistent and asymptotically efficient, while the fixed effects ($\beta^{EF}$) are only consistent. In this case, the random effects estimator is chosen. If $H_0$ is rejected, evidencing that $E(a|X)\neq0$, random effects become inconsistent, and the fixed effects estimator is preferred.

The robustness of the models is analyzed through the detection of spatial dependence using Lagrange Multiplier (LM) tests. These tests evaluate the significance of spatial effects in econometric models, being fundamental to determine whether spatial dependence should be included in the model [21]. The tests are:

- **LML:** Tests the hypothesis that spatial dependence is in the dependent variable, suggesting the adoption of a SAR model. The statistic is given by: $LM_{Lag} = \frac{(e'We)^2}{a^2 \cdot tr(W'W)}$, Where e are the model residuals, W is the spatial weight matrix, and a2 is the residual variance.

- **LME:** Tests the hypothesis that spatial dependence is in the errors, suggesting the adoption of a SEM model. The statistic is calculated by:: $LM_{Error} = \frac{e'WWe}{a^2}$.

- **Robust-RLM e Robust-LME:** These are adjusted versions of the LML and LME tests, considering the presence of other forms of spatial dependence. The robust statistics are adjusted to control for the influence of spatial error in the lag test and vice versa. They are expressed as: $RLM_{Lag} = \frac{RLM_{Lag}}{1-\frac{1}{T}.tr(W'W)}$ e $RLM_{Error} = \frac{RLM_{Error}}{1-\frac{1}{T}.tr(WW')}$. Where T is the number of time observations.

The interpretation is based on the statistical significance of the obtained values. If the values are significantly different from zero (p<0.05p < 0.05p<0.05), this indicates the presence of spatial autocorrelation, either global or local, depending on the applied test. This evidence suggests the need to use specific models, such as SAR, SEM, or SDM, that can better capture the spatial structure of the data [21].

To compare the SDM specification with SAR and SEM, two null hypotheses are tested: $H_0^1$ : $Y = 0$ and $H_0^2 : Y = -\beta p$. Validation $H_0^1$ suggests that a SAR model would better fit the data, while validation of $H_0^2$ indicates that a SEM model is more appropriate. Simultaneous rejection of $H_0^1$ and $H_0^2$ leads to the selection of the SDM model [21].

In addition to LM tests, the likelihood value and R-squared ($R^2$) are also used to evaluate the model's fit to the observed data. The likelihood statistic seeks to maximize its value, while the $R^2$ indicates the proportion of the total variability of the dependent variable explained by the model. The higher the $R^2$, the better the model fit, providing a comprehensive metric to assess the model's performance in explaining the observed variability [65].

## Data description

The research sample is composed of all municipalities in Brazil (5,570), organized into a balanced panel covering four years (2018–2021), totaling 22,280 observations. The data were collected from various official sources, including the General Register of Employed and Unemployed (CAGED), the Annual Social Information Report (RAIS), the Secretariat for Evaluation and Information Management (SAGI), the Brazilian Institute of Geography and Statistics (IBGE), the Unified Health System (COVID-19 Monitoring Panel), and the Integrated Disaster Information System (S2id). These sources are widely used in studies due to their comprehensiveness, methodological transparency, and historical continuity, ensuring a high level of reliability for the proposed analysis. Table 1 displays the variables used in the econometric analyses.

**Table 1. Presentation, description, and source of the variables used in this study.**

| Code | Variable | Description | Source |
|---|---|---|---|
| **y** | Total hires | The ratio of workers hired in the agricultural sector during the year to the total number hired in all sectors in the same year (%). | CAGED |
| $x_1$ | Wages | The sum of the wages of people employed in the agricultural sector in the municipality during the reference year (in millions BRL). | CAGED |
| $x_2$ | Income | The ratio between the wage bill (in millions BRL) of employees in the agricultural sector and the total wage bill of all other sectors (%). | RAIS |
| $x_3$ | Bolsa Família | Total amount passed on to Brazilian municipalities divided by the number of beneficiary families in the Bolsa Família Program/Auxílio Brasil Program (Bolsa Família in force until 10/2021) and Auxílio Brasil (11 and 12/2021) (in thousands BRL). | SAGI |
| $x_4$ | CCB | Continuous Cash Benefit per municipality.–Mean. Obtained by summing the number of disabled and elderly individuals receiving the CCB by municipality. The total value of CCB by municipality is divided by the number of beneficiaries (in thousands BRL). | SAGI |
| $x_5$ | Emergency Aid | Total amount (sum of all publics) referring to Emergency Aid per municipality, which includes the following recipients: total amount to be transferred to the Cadastro Único (Unique Registry) recipients, total amount to be transferred to the Extracad recipients, total amount to be transferred to the Judicial recipients, and total amount to be transferred to the Bolsa Família recipients (in millions BRL). | SAGI |
| $x_6$ | COVID-19 cases | The proportion of COVID-19 cases per municipality population. Obtained by dividing the number of COVID-19 cases by the population of each municipality and multiplying by one thousand. | Control Panel (COVID) |
| $x_7$ | COVID-19 deaths | The proportion of COVID-19 deaths per municipality population. Obtained by dividing the number of COVID-19 deaths by the population of each municipality and multiplying by one thousand. | Control Panel (COVID) |
| $x_8$ | Extreme events | Dummy variable for extreme events, including droughts, dry spells, frosts, and floods. Assumes a value of 1 when the municipality was affected by one of these events and 0 if there was no occurrence. | S2id |
| $x_9$ | Demographic density | Expresses the ratio between the population of each municipality and the surface area of the territory. Population density is obtained from the ratio between the total number of inhabitants per municipality and the total area (%). | IBGE |

Source: IBGE, RAIS, CAGED, SAGI, S2id, COVID-19 Control Panel.

The choice of the time frame from 2018 to 2021 was made to capture both pre-existing effects and the direct impact of the COVID-19 pandemic on the agricultural sector and municipal assistance policies, providing a baseline that allows for the identification of the economic and social conditions immediately prior to the pandemic. This timeframe facilitates the comparison and distinction of effects directly caused by the health and economic crisis of 2020 and 2021, something that would be challenging without a comparable baseline period.

The exclusion of periods prior to 2018 was intended to avoid possible distortions in the analysis, as events such as the economic crisis of 2015–2016 also significantly affected the agricultural sector and rural employment [66, 67]. The temporal limitation to 2021 is related to the availability of data collected from official sources up to the time of the analysis. Furthermore, the cutoff in 2021 reflects the phase during which emergency assistance policies were most intensively applied, such as the Emergency Aid program, whose distribution was largely discontinued after the end of 2021.

Regarding data treatment, it was found that there was a low proportion of missing values, not exceeding 3% in any variable. In this context, the simple median imputation technique was adopted, as recommended by Harrell [68]. This approach is recommended in situations of low data absence, as it is less sensitive to outliers than the mean, avoiding distortions in the results of econometric analyses. This method is also indicated when aiming to avoid the loss of statistical power that could occur with other techniques, such as complete case exclusion (listwise deletion), which can reduce sample size and the precision of econometric models [69].

### Heterogeneity and robustness tests

We adopted a heterogeneity analysis to examine regional variations in the impacts of the COVID-19 pandemic on agricultural employment, segmenting the sample into the five major regions of Brazil (see Fig 1). This strategy aims to capture potential structural differences in the effects of the explanatory variables, considering regional disparities. Previous studies indicate that the pandemic affected different geographical areas in varied ways, influenced by specific economic, social, and infrastructure characteristics of each region [70, 71]. Thus, this approach allows for the identification of disparities in the impact of COVID-19 on the agricultural sector that could be masked in a national aggregated analysis.

In addition to the heterogeneity analysis, the consistency of results is verified through robustness tests. This approach aims to ensure the validity and reliability of the results by analyzing the significance and consistency of the effects of explanatory variables in different specifications [72]. We propose to test the consistency and significance of the effects of COVID-19-related variables by substituting the dependent variable. To do this, we use the ratio of wages paid to agricultural workers compared to the wages paid to the total number of workers across all economic sectors. The goal is to assess how the relative participation of agricultural wages was affected by the pandemic. By doing this, we provide an additional and broader perspective on the economic impact of COVID-19.

This strategy aims to strengthen the robustness of the analysis by demonstrating that the results are not sensitive to the choice of a single dependent variable. The consistency of the findings, even when shifting from an analysis focused solely on employment to one that considers the relative participation of wage mass, reinforces that the observed effects are robust and reflect real impacts.

## Results and discussion

The examination of the findings is structured into three distinct sections. The initial section is dedicated to the analysis of the descriptive statistics—specifically, the mean—of all variables across various periods spanning from 2018 to 2021. The subsequent section delves into the spatial analysis of the data, focusing primarily on the dependent variable through the application of Moran's I. Lastly, the econometric outcomes derived from panel data analyses are discussed.

### Model statistics

Table 2 presents the annual average results for the variables included in the model. Throughout the period under review, a noticeable decline was observed in the percentage of individuals employed in agricultural positions relative to the total number of individuals employed, with this proportion reaching its lowest point of 13.70% in 2021. This metric experienced a precipitous decline in the inaugural year of the pandemic, in contrast to trends observed in other labor market sectors. The count of municipalities identified for employing agricultural workers decreased from 4,806 in 2018 to 3,700 in 2020. However, this downturn was not mirrored in terms of compensation levels during the pandemic; the annual average aggregate wages of those employed in the agricultural sector exhibited a marked increase, particularly in 2020 and 2021 (Table 2).

The geographic pattern of the ratio of workers admitted to the agricultural sector compared to the total admitted in all sectors can be observed in the maps presented in Fig 3. These maps illustrate the distribution of the percentage of admissions disaggregated by quantiles in Brazilian municipalities during the period from 2018 to 2021 (upper maps). The delta map, located below the absolute maps, provides an overview of the changes that occurred between these two

**Table 2. Variables used in the spatial regression model (annual averages).**

| Variables | 2018 | 2019 | 2020 | 2021 |
|---|---|---|---|---|
| Total hires | 25.8 | 25.25 | 15.53 | 13.7 |
| Agricultural Salary/Total Salary | 0,241 | 0,237 | 0,211 | 0,235 |
| Wages | 256,086.60 | 262,542.50 | 1,835,582.00 | 1,819,098.60 |
| Income | 9.50% | 9.44% | 9.78% | 9.44% |
| Bolsa Família | 2,096.14 | 2,159.82 | 706.56 | 1,569.11 |
| CCB | 7,878.73 | 6,330.17 | 941.26 | 1,047.41 |
| Emergency Aid | 0 | 0 | 68,000,000.00 | 91,629.57 |
| COVID-19 cases | 0 | 0 | 76.16 | 72.67362 |
| COVID-19 deaths | 0 | 0 | 34.95 | 1.673553 |
| Extreme events | 0.22 | 0.21 | 0.3 | 0.26 |
| Population density | 118.65 | 119.63 | 120.6 | 121.49 |

Source: prepared by the authors based on data from IBGE, and RAIS, CAGED, SAGI, S2id, COVID-19 Control Panel.

periods. It highlights the variations in the ratio of agricultural workers, demonstrating where and to what extent these changes took place. This representation aids in understanding not only the static distribution of workers but also the temporal dynamics and the effects of policies or interventions over time.

The colors indicate the magnitude of admission levels in the agricultural sector, with municipalities highlighted in yellow belonging to the highest quantile ($Q_4$), representing those responsible for the largest proportion of admitted agricultural workers. It can be observed that the Southeast and Central-West regions of Brazil accounted for the highest volume of admissions, while municipalities located in the North and Northeast regions, highlighted in dark colors ($Q_1$), exhibit lower rates.

Based on the lower map, it is evident that the most significant changes in agricultural employment patterns during this period occurred particularly in the North (especially in the state of Amazonas) and Northeast regions, where admission rates are lower compared to the other regions. These regions experienced a substantial shift in the profile of agricultural admissions between the two analyzed periods. The state of Amazonas, for instance, faced extreme challenges during the pandemic, including the collapse of the healthcare system in Manaus [73]. This collapse, combined with the exponential increase in deaths and the public health crisis, may have led to a significant displacement of workers into the agricultural sector, seeking safer alternatives outside of overcrowded urban centers.

## Moran's index

The preliminary analysis aims to ascertain through statistical examination whether the spatial distribution of the data can be attributed to randomness. Specifically, it endeavors to determine if a variable's value in a given municipality or region is influenced by the values in neighboring localities [59]. This analysis is crucial for developing indicators that validate the application of spatial econometric models. The existence of a correlation between observations signifies a violation of the independence assumption, which is a core prerequisite for the classical linear regression model estimated through ordinary least squares. Such a violation results in biased and inefficient estimators [72].

In the realm of spatial data analysis, the global Moran's index is frequently employed as a statistic to explore the spatial dependence of data. This statistical tool is utilized to investigate the presence of a systematic spatial structure in the observations, thereby indicating spatial

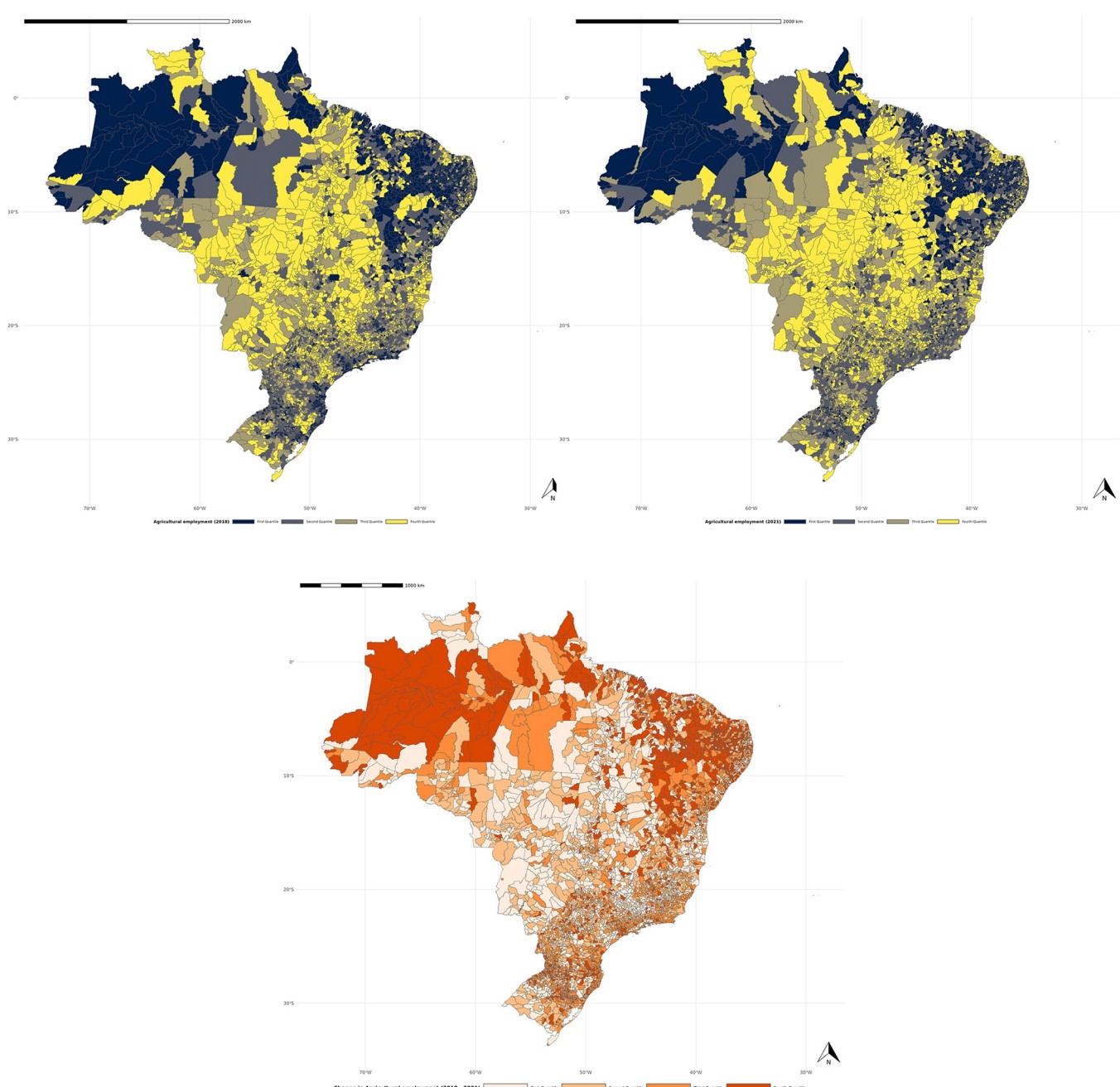

**Fig 3. Total hiring ratio for 2018 and 2021 (in quartiles).** Source: The map was created using the ggplot2 and geobr packages available for the R software. The data plotted on the map were obtained from the General Register of Employed and Unemployed (CAGED), provided by the Ministry of Labor and Employment (MTE). All data used are publicly available and compatible with the CC BY 4.0 license.

autocorrelation [74]. The outcomes of Moran's I indicator for spatial autocorrelation throughout the study are presented in Table 3.

For all observed periods, the significance levels were at 1%, demonstrating that the proportion of individuals employed in the agricultural sector relative to the overall labor market did not arise by chance. Instead, it resulted from spatial factors that drive geographical clustering among neighboring municipalities. Furthermore, it is crucial to acknowledge that, despite the

**Table 3. Moran's I–the ratio between the hires in the agricultural sector in 2018–2021.**

| Year | Moran's I | Z |
|---|---|---|
| 2018 | 0.367*** | 45.086 |
| 2019 | 0.358*** | 44.001 |
| 2020 | 0.236*** | 29.039 |
| 2021 | 0.260*** | 32.004 |

Note
*$p < 0.1$
**$p < 0.05$
***$p < 0.01$.
Source: Research data.

indicator's downward trend throughout the period, it reveals the presence of positive spatial autocorrelation. This initial observation underscores the importance of considering spatial correlation to gain a more accurate and holistic insight into the patterns and processes associated with this phenomenon.

## Panel data models

The identification of the spatial process through which the analyzed phenomenon demonstrates spatial autocorrelation, whether through the lag of the dependent variable or spatial dependence in residual form, is conducted using the Lagrange Multiplier for Lag (LML) test, the Lagrange Multiplier for Error (LME) test, and their robust variants, the Robust Lagrange Multiplier for Lag (RLML) test and the Robust Lagrange Multiplier for Error (RLME) test (Table 4).

The results of the spatial dependence tests indicate statistical significance in all the tested models ($p < 0.01$), confirming the presence of spatial autocorrelation in both the dependent variable and the residuals. The rejection of the null hypothesis (H$_0$) in all tests suggests that the SDM (Spatial Durbin Model), which combines elements of the SAR (Spatial Autoregressive) and SEM (Spatial Error Model) models by incorporating spatial dependence in both the dependent variable and the error term, may be more suitable for capturing the spatial structures of the data [21]. This specification provides a more robust solution for investigating the direct and indirect effects of explanatory variables on the dependent variable.

The analysis of the log-likelihood (see S1 Table) reinforces the choice of the SDM as the most appropriate model. The SDM presented a higher log-likelihood value (-82,818.81),

**Table 4. Diagnostic test for the spatial specifications.**

| | Pooled OLS | Spatial fixed effects | Time fixed effects | Spatial and time fixed effects | Random effects |
|---|---|---|---|---|---|
| LML | 4151.70*** | 2260.33*** | 3997.54*** | 1766.43*** | 3217.33*** |
| LME | 5044.61*** | 2735.77*** | 4693.53*** | 1722.61*** | 3762.17*** |
| Robust LML | 90.59*** | 246.26*** | 71.94*** | 48.38*** | 151.31*** |
| Robust LME | 983.49*** | 721.71*** | 721.71*** | 721.71*** | 696.15*** |

Note
*$p < 0.1$
**$p < 0.05$
***$p < 0.01$.
Source: Research data.

**Table 5. The results of the spatial Hausman test.**

| Verifications | $\chi 2$ | DF | $p$ |
|---|---|---|---|
| SAR EF (individual) vs. SAR EA | 395.37 | 9 | < 0.000 |
| SAR EF (time) vs. SAR EA | 2027.00 | 9 | < 0.000 |
| SAR EF (two ways) vs. SAR EA | 649.57 | 9 | < 0.000 |
| SEM EF (individual) vs. SEM EA | 869.39 | 9 | < 0.000 |
| SEM EF (time) vs. SEM EA | 132.31 | 9 | < 0.000 |
| SEM EF (two ways) vs. SEM EA | 475.22 | 9 | < 0.000 |
| SDM EF (individual) vs. SDM EA | 506.97 | 18 | < 0.000 |
| SDM EF (time) vs. SDM EA | 296.00 | 18 | < 0.000 |
| SDM EF (two ways) vs. SDM EA | 506.05 | 18 | < 0.000 |

Source: Research data.

indicating a greater ability to explain the variability of the dependent variable by maximizing the likelihood compared to other models tested [65]. Furthermore, the higher coefficient of determination ($R^2$) for the SDM strengthens its robustness, showing that 81.79% of the variation in agricultural employment across Brazilian municipalities is explained by the independent variables used.

After determining the most appropriate spatial specification, the Hauman test is applied (Table 5) to each pair of models to identify potential differences between the coefficients of the consistent estimator (Fixed Effects) and the efficient estimator (Random Effects). The goal now is to identify the best estimation method within the spatial panel structure.

Upon reviewing the outcomes derived from tests assessing spatial patterns (Table 4), the Hausman test (Table 5), as well as the log-likelihood and $R^2$ estimates (S1 Table), it is posited that the SDM specification, which incorporates both spatial and temporal fixed effects, provides a superior fit for the data compared to the SAR and SEM specifications (Table 6).

The significant and positive estimate of the spatially lagged term λ underscores the importance of considering the spatial dynamics among neighboring municipalities when analyzing employment trends in the agricultural sector in Brazil. In particular, the value of λ suggests that a 1% increase in the proportion of individuals employed in the agricultural sector relative to the total employment in neighboring municipalities leads to a mean increase of 0.34% in these municipalities.

The variables introduced to test aspects related to both agricultural income and income from social programs were found to be significant, except for the BPC variable. As expected, income generated in the sector positively impacts employment, although the magnitude of the impact is low given the coefficient value. However, when observing the signs of the spatially lagged coefficients (W_Agro Salary; W_Agro Income), a negative signal is observed, indicating that an increase in income in one municipality generates an average negative effect on neighboring municipalities.

In addition to agricultural income, income from social programs implemented by the Federal Government, such as Bolsa Família and Emergency Aid, showed positive effects on those employed in the agricultural sector. This means that an increase in the transfers from these programs did not lead to a decline in the hiring of workers in the studied sector. This effect indicates that Emergency Aid not only helped sustain the purchasing power of the most vulnerable population, as highlighted by Viollaz *et al*. [75] and Khamis *et al*. [24], but also brought positive impacts for employment in the agricultural sector. Studies such as Souza *et al*. [19] reinforce that the rapid increase in transfers was essential for financial inclusion and

**Table 6.** *The spatial and temporal fixed effects for the ratio of people hired in the agricultural sector in relation to the total number of people hired.*

| Variables | Coefficient (SE) | Variables (W) | Coefficient (SE) |
|---|---|---|---|
| $\lambda$ | 0.3488*** | | |
| Wages (x1) | 7.3427e-07*** (5.3732e-08) | W_Wage (wx1) | -1.1854e-06*** (9.9841e-08) |
| Income (x2) | 9.2043e-07** (2.8215e-07) | W_Income (wx2) | -3.6511e-06*** (5.3986e-07) |
| Bolsa Família (x3) | 2.6481e-03*** (5.3837e-04) | W_Emergency Aid (wx3) | -4.5652e-03*** (7.6820e-04) |
| CCB (x4) | -1.2169e-05 (2.6840e-05) | W_CCB (wx4) | 1.4493e-04** (5.5986e-05) |
| Emergency Aid (x5) | 8.9709e-03*** (2.3275e-03) | W_ Emergency Aid (wx5) | 9.7058e-04 (2.3683e-03) |
| COVID-19 cases (x6) | 1.5852e-04*** (2.2109e-05) | W_ COVID-19 cases (wx6) | -3.4568e-05 (3.9655e-05) |
| COVID-19 deaths (x7) | -3.0466e-03*** (5.4077e-04) | W_COVID-19 cases (wx7) | 9.8497e-04 (9.3914e-04) |
| Extreme events (x8) | -3.8093e-01 (3.4936e-01) | W_ Extreme events (wx8) | 6.0545e-01 (5.0980e-01) |
| Population density (x9) | 3.3205e-02* (1.5347e-02) | W_Demographic density (wx9) | 1.2013e-01*** (2.7023e-02) |

SE: Standard error

*$p < 0.1$

**$p < 0.05$

***$p < 0.01$. Source: prepared by the authors based on research data.

protection of the most vulnerable, helping to prevent drastic job losses in both formal and informal sectors.

However, it was observed that the increase in Bolsa Família generated a negative spatial effect on neighboring municipalities, suggesting that an increase in transfers in a given municipality may lead to a proportional decrease in the hiring of workers in the agricultural sector in adjacent regions. This phenomenon may be related to a growing dependency on government aid, discouraging work in neighboring municipalities, or even limited worker mobility during the pandemic. Lazzari *et al.* [20] also highlight that the impact of benefits varied across different occupational groups, which may explain the complex spatial dynamics observed in the agricultural sector, with the most vulnerable workers being the most benefited by the policy, while less favored regions felt the negative impacts of these transfers in terms of employment.

The variables introduced to measure the effects of the pandemic, such as the proportion of COVID-19 cases and deaths, proved significant in explaining variations in agricultural employment. The variable representing the proportion of cases relative to the population showed a positive effect on employment, supporting the literature that suggests the agricultural sector played an important role in absorbing labor displaced from sectors such as industry and services during the crisis [25, 26]. This migration occurred partly due to the essential nature of agricultural activities, which continued operating even with the severe restrictions imposed by the pandemic [22–24] also emphasize that the sector was less affected compared to others, absorbing workers who lost their jobs in cities.

Conversely, the death variable had a negative effect, indicating that the increase in deaths adversely impacted the hiring of workers in the sector. This result supports the view that,

despite the resilience of the agricultural sector, the severity of the pandemic, as measured by mortality, posed significant challenges. The reduction in the available workforce, especially in areas with higher mortality rates, impacted agricultural production. This finding aligns with studies like those by Zabir *et al.* [7] and Kakaei *et al.* [8], which highlighted that disruptions in supply chains and the loss of seasonal workers due to the pandemic were key factors contributing to the decline in activity in some regions.

However, the spatial effects (W) of the variables related to the pandemic were not statistically significant. This may indicate that while the direct impacts on municipalities are evident, the effects on neighboring areas did not propagate substantially. In other words, while a municipality with a high number of cases or deaths may face local employment challenges in agriculture, these difficulties do not seem to significantly affect neighboring regions. Studies like Yao *et al.* [35], which analyzed the geographically weighted effects of the pandemic, suggest that these impacts are generally localized, without widespread propagation to surrounding areas, aligning with the results found in this study at the national level.

The lack of significance of the variable representing the occurrence of extreme weather events may be attributed to several causes. Firstly, the spatial and temporal variability of the impacts of extreme weather events can dilute their direct and indirect effects on employment. In some regions, farmers may adopt mitigation strategies, such as crop diversification, irrigation, or precision technology, thereby reducing reliance on labor during critical periods [40]. Additionally, the effects of these events may not be immediate, with agricultural losses being felt only in subsequent cycles, making it difficult to capture their direct influence in the short term.

Another possible factor is that, although extreme weather events affect production, the demand for labor may be partially compensated by the need for recovery in affected areas, such as rebuilding agricultural infrastructures or replanting, which would keep employment levels relatively stable [43]. Finally, the lack of significance may also indicate the resilience of certain productive chains in Brazil, which can sustain high levels of employment in the face of climatic challenges, thanks to government support and the growing adoption of agricultural insurance and production support policies. [41]. These factors may explain the absence of a clear and consistent impact of extreme weather events on agricultural employment, both directly and indirectly.

## Heterogeneity and robustness tests

Table 7 presents the results of models focusing on the direct and indirect effects of pandemic-related variables (COVID-19 Cases and Deaths) on agricultural employment across major Brazilian regions. The heterogeneity analysis by region seeks to identify differentiated patterns in the impacts of the pandemic based on regional characteristics.

The analysis of the impacts of COVID-19 on agricultural employment reveals distinct regional patterns, demonstrating that the effects of the pandemic were not homogeneous at the national level. In the North region, for example, it was observed that the impact of cases was positive and greater than the national average, suggesting that the agricultural sector managed to maintain its operations, possibly due to its representativeness and essential nature in the region [47]. However, the negative effect of deaths may reflect greater fragility in the healthcare infrastructure. In states like Amazonas, for instance, the healthcare system collapsed, with an exponential increase in deaths [73], amplifying the vulnerability of the local workforce.

Similarly to the North region, the Northeast of Brazil exhibits patterns of interdependence among municipalities, with contrasting results for cases and deaths when considering indirect

**Table 7. Direct and indirect effects of COVID19 on agricultural employment in Brazilian regions.**

| Region | Covid Cases | W | Covid Deaths | W | λ |
|---|---|---|---|---|---|
| North | 0.0003** (0.00014) | -0.0019*** (0.0002) | -0.0064* (0.0038) | 0.0421*** (0.0070) | 0,4474*** (0,0271) |
| North East | 0.00013 (0.00010) | -0.0013*** (0.0001) | -0.0036 (0.0033) | 0.03514*** (0.0049) | 0,3308*** (0,0155) |
| Midwest | 0.000006 (0.00009) | -0.000445 (0.00020) | 0.00159 (0.00411) | 0.00910* (0.00818) | 0,6642*** (0,0212) |
| Southeast | 0.00007** (0.00003) | -0.00024*** (0.00004) | -0.0012* (0.00072) | 0.0052*** (0.00117) | 0.6543*** (0.01189) |
| South | 0.0002*** (0.00006) | -0.00060*** (0.000088) | -0.0054** (0.0021) | 0.01536*** (0.0032) | 0.5921*** (0.0150) |

SE: Standard error

*$p < 0.1$

**$p < 0.05$

***$p < 0.01$. Source: prepared by the authors based on research data.

effects (W). In both regions, an increase in the number of cases in one municipality leads to a decrease in agricultural employment in neighboring municipalities. This decline may be associated with a perception of risk, with workers avoiding travel and employers reducing their operations due to uncertainty. On the other hand, an increase in the number of deaths in a municipality seems to have the opposite effect, leading to an increase in agricultural employment in neighboring municipalities. One hypothesis is that the rise in deaths creates a need to replace workers in areas directly impacted, prompting neighboring municipalities to absorb this demand.

In the Southeast, although the direct impact of cases was positive, it was lower than in the North. The more mechanized and technologically advanced structure of the agricultural sector in the region may have influenced this difference [53, 54]. Additionally, the direct impact of deaths was negative but less so than in the North and South. The broader healthcare infrastructure may have mitigated adverse impacts. Similarly, the Central-West region, with its higher level of mechanization and economy heavily reliant on agribusiness, particularly focused on commodity exports, showed greater resilience. [52]. Both COVID-19 cases and deaths had a limited impact on agricultural employment, suggesting that the pandemic did not significantly interfere with the sector compared to other regions of the country. Spatial dependence presented the highest index among the regions (λ: 0.6642), reflecting the importance of the sector to the local economy.

In the South, the impact of cases was positive, indicating an increase in agricultural demand. This result may be related to the relevance of crops like soy in this region, which experienced record export increases during this period [28]. Conversely, mortality had a considerable negative impact, possibly reflecting a greater reliance on seasonal labor in agricultural production, affected by deaths resulting from the pandemic. Agricultural production in the South significantly depends on seasonal labor, especially for activities like harvesting and the cultivation of certain products such as tobacco, grapes, rice, and vegetables [55].

In addition to the heterogeneity tests, a robustness test was conducted by introducing a new dependent variable that represents the ratio of agricultural wage mass to the total wages paid. This change allows for verifying whether the identified impacts remain consistent when evaluating the agricultural sector from another economic perspective. Table 8 presents the results of the model.

**Table 8. Direct and indirect effects of COVID-19 on agricultural wages.**

| Variables | Coefficients | w |
|---|---|---|
| Covid Cases | 0.000001*** (0.0000004) | 0.000001* (0.00000071) |
| Covid Deaths | -0.000038*** (0.000010) | -0.000026 (0.0000174) |
| $\lambda$ | 0.17035*** (0.00996) | |

SE: Standard error

*$p < 0.1$

**$p < 0.05$

***$p < 0.01$. Source: prepared by the authors based on research data.

Based on the obtained results, the robustness of the model is confirmed. The lambda coefficient ($\lambda$) remains significant, indicating that the spatial structure of the data is relevant, regardless of the dependent variable used. Additionally, both in the original model and in the robustness test, the effects of variables related to COVID-19 (cases and deaths) remain significant and with consistent signs, indicating that even when the focus shifts to wage mass, the effects of the pandemic remain clear: more cases of COVID-19 correlate with a greater share of the agricultural sector, likely due to its essential nature during the pandemic.

## Conclusions

This study aimed to understand the impacts of multiple dimensions on employment in the Brazilian agricultural sector, investigating how different factors, including public policies, climatic events, and the pandemic itself, influenced this dynamic. From the analyses conducted, it is possible to discuss the confirmation or rejection of the raised hypotheses and reflect on the contributions and limitations of the study.

The empirical analysis revealed that the proportion of COVID-19 cases had a positive effect on agricultural employment. This result corroborates the first hypothesis raised ($H_1$), reinforcing evidence that the agricultural sector absorbed workers displaced from other sectors during the crisis. Considering the heterogeneity tests, it is inferred that this absorption was more intense among workers from the same municipality, as the indirect effects were negative in all regions. The results highlight the resilience of the agricultural sector, which continued to operate even amidst restrictions, helping to maintain employment during a period of great economic uncertainty.

The second hypothesis ($H_2$), which suggests that the impacts of COVID-19 present spatial spillover effects on agricultural employment, was only partially confirmed. Although no significant effects were observed at the national level, the heterogeneity test revealed that in the North, Northeast, Southeast, and South regions, the indirect effects, both for cases and deaths, are significant. The effects for the proportion of cases suggest that the higher the number in a municipality, the lower the agricultural employment in neighboring municipalities, possibly due to the perceived risk associated with the pandemic. The lack of significance observed at the national level may be attributed to the greater resilience of the Central-West region, the country's prime main agricultural area. With its more mechanized sector, this region experienced less impact compared to others. This resilience may have mitigated the spatial spillover effect, resulting in the absence of substantial propagation at the national level.

The analysis did not confirm the third hypothesis raised ($H_3$), revealing that extreme climatic events did not significantly impact agricultural employment during the period. The spatial and temporal variability of climatic impacts may have influenced the lack of significance. In some regions, these events may be more frequent or intense, while in others, the impacts may be less pronounced or even null. The variability in the magnitude and distribution of these events may dilute the observable impact on agricultural employment, making it difficult to capture a consistent effect at an aggregated level.

The fourth hypothesis was partially confirmed ($H_4$); social programs, such as Bolsa Família and Auxílio Emergencial, had a positive effect on agricultural employment, supporting the maintenance of hiring levels in the sector. This finding reinforces the importance of public policies in protecting jobs and sustaining income during the crisis. However, a negative effect was also observed in neighboring municipalities, suggesting possible undesirable effects associated with increased transfers when considering adjacent regions.

The impacts of the pandemic on agricultural employment in Brazil can be considered heterogeneous, confirming the fifth hypothesis ($H_5$). Regional differences, such as the resilience of agribusiness in the Central-West and the vulnerability of agricultural employment in the North and South, highlight that the mitigation policies of COVID-19 and the socioeconomic effects of the pandemic varied across regions. The analysis suggests that any formulation of public policies should consider these regional disparities to ensure more effective and targeted responses to the specific challenges faced by each region.

The study contributes to the literature by addressing the determinants of agricultural employment in the context of a severe crisis using a spatial approach, which is relatively rare in the existing literature. By applying models that capture spatial interdependencies, the study reveals how the effects of the pandemic spread and influenced neighboring areas, providing a more refined understanding of regional dynamics. Additionally, although the focus was on Brazilian municipalities, the findings may be relevant to other developing countries with significant agricultural economies and similar social structures. In contexts where the agricultural sector plays an essential role, similar patterns of resilience may be observed, particularly in areas where seasonal employment and exports are important.

However, it is important to note that each country's particularities—such as the structure of production chains, specific public policies, and the degree of agricultural mechanization—affect how these factors interact with local employment. Therefore, while the results of this research contribute to the debate on the impact of COVID-19 on the agricultural sector, their applicability to other contexts requires careful consideration of each country's economic and social specificities.

The prospects for future research can be expanded in several directions. First, it is suggested to investigate how different crops and agricultural products were specifically impacted, taking into account the unique characteristics of each product and the demand fluctuations during the pandemic. This could provide a more granular view of the economic impacts on the sector. Additionally, it would be valuable to include additional variables that capture access to technology and education, factors that can directly influence agricultural workers' ability to adapt to changing employment conditions. The level of mechanization, for instance, may be a crucial determinant of the sector's resilience in the face of crises.

With the availability of more post-pandemic data, it would be interesting to extend the analysis period to investigate the long-term effects, including the recovery or stagnation of agricultural employment in different regions. Moreover, expanding the geographical scale, considering comparisons with other agricultural countries, would allow for a more comprehensive analysis of the pandemic's global effects on employment in the sector.

## Supporting information

**S1 Table. Model estimates in different spatial patterns.** ***$p < 0.001$; **$p < 0.01$; *$p < 0.05$; . $p < 0.1$. Source: Research data.
(DOCX)

**S1 Dataset. Dataset used in the research.** The variables are coded according to Table 1 of the manuscript. In addition to the variables used in the models, the panel data contains additional information representing the ID of each municipality (id), the municipality code (codmun), the name of the municipality (municipality), the state abbreviation (state_abr), and the year of observation (year).
(XLSX)

## Acknowledgments

We thank the Austral University of Chile and the Pontifical Catholic University of Rio Grande do Sul (Brazil) for their unconditional support for scientific research.

## Author Contributions

**Conceptualization:** Patrícia Batistella, Fernanda Castilhos França de Vasconcellos, Augusto Mussi Alvim, Adelar Fochezatto.

**Data curation:** Patrícia Batistella, Luan Marca, Fernanda Castilhos França de Vasconcellos.

**Formal analysis:** Patrícia Batistella, Luan Marca, Augusto Mussi Alvim, Adelar Fochezatto.

**Funding acquisition:** Eduardo Rodrigues Sanguinet.

**Investigation:** Patrícia Batistella, Luan Marca, Fernanda Castilhos França de Vasconcellos, Eduardo Rodrigues Sanguinet, Augusto Mussi Alvim, Adelar Fochezatto.

**Methodology:** Patrícia Batistella, Augusto Mussi Alvim, Adelar Fochezatto.

**Project administration:** Eduardo Rodrigues Sanguinet, Augusto Mussi Alvim.

**Resources:** Patrícia Batistella, Luan Marca.

**Supervision:** Eduardo Rodrigues Sanguinet, Augusto Mussi Alvim.

**Validation:** Patrícia Batistella, Luan Marca, Augusto Mussi Alvim.

**Visualization:** Patrícia Batistella, Augusto Mussi Alvim, Adelar Fochezatto.

**Writing – original draft:** Patrícia Batistella, Luan Marca, Fernanda Castilhos França de Vasconcellos, Eduardo Rodrigues Sanguinet, Augusto Mussi Alvim.

**Writing – review & editing:** Luan Marca, Augusto Mussi Alvim.

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
