## [Decision Letter · Decision Letter 0]

26 Aug 2024

PONE-D-24-24987The Effects of the COVID-19 Pandemic on Agricultural Employment in Brazilian Municipalities

PLOS ONE

Dear Dr. Augusto Alvim Mussi,

Thank you for submitting your manuscript to PLOS ONE. After careful consideration, we feel that it has merit but does not fully meet PLOS ONE’s publication criteria as it currently stands. Therefore, we invite you to submit a revised version of the manuscript that addresses the points raised during the review process.

Your study on the impact of the COVID-19 pandemic on agricultural employment in Brazil, using spatial econometric models, addresses an important and timely issue. However, before your paper can be considered for publication, substantial revisions are needed.

First, the research question needs clarification, particularly in broadening the focus beyond employment to encompass the overall impact on agriculture in Brazil. Additionally, providing an overview of the current state of agricultural development in Brazil would help set the context for your study. In the literature review, it is important to discuss comparative studies on the pandemic’s impact on agricultural employment, as well as research on the topic that uses spatial econometric models. Organizing the review by themes could improve its structure and coherence.

The methodology section requires more detail, especially regarding the construction and rationale of the spatial weight matrix, along with the steps and results of model selection. For the data section, please elaborate on the methods of data collection, the reliability of your sources, and how missing data was handled. Lastly, it is essential to include robustness tests to validate your model selection.

We look forward to receiving your revised manuscript.

Kind regards,

Carolina Serpieri, PhD

Academic Editor

PLOS ONE

Journal Requirements:

When submitting your revision, we need you to address these additional requirements. 1. Please ensure that your manuscript meets PLOS ONE's style requirements, including those for file naming. The PLOS ONE style templates can be found at https://journals.plos.org/plosone/s/file?id=wjVg/PLOSOne_formatting_sample_main_body.pdf and https://journals.plos.org/plosone/s/file?id=ba62/PLOSOne_formatting_sample_title_authors_affiliations.pdf 2. We note that the grant information you provided in the ‘Funding Information’ and ‘Financial Disclosure’ sections do not match.  When you resubmit, please ensure that you provide the correct grant numbers for the awards you received for your study in the ‘Funding Information’ section. 3. Thank you for stating the following financial disclosure: "Agencia Nacional de Investigacion y Desarrollo" Please state what role the funders took in the study.  If the funders had no role, please state: ""The funders had no role in study design, data collection and analysis, decision to publish, or preparation of the manuscript."" If this statement is not correct you must amend it as needed. Please include this amended Role of Funder statement in your cover letter; we will change the online submission form on your behalf. 4. We note you have included a table to which you do not refer in the text of your manuscript. Please ensure that you refer to Table 7 in your text; if accepted, production will need this reference to link the reader to the Table. 5. We notice that your supplementary tables are included in the manuscript file. Please remove them and upload them with the file type 'Supporting Information'. Please ensure that each Supporting Information file has a legend listed in the manuscript after the references list.

Reviewers' comments:

Reviewer's Responses to Questions

**Comments to the Author**

1. Is the manuscript technically sound, and do the data support the conclusions?

Reviewer #1: Partly

Reviewer #2: Partly

Reviewer #3: Yes

2. Has the statistical analysis been performed appropriately and rigorously? 

Reviewer #1: No

Reviewer #2: Yes

Reviewer #3: Yes

3. Have the authors made all data underlying the findings in their manuscript fully available?

Reviewer #1: Yes

Reviewer #2: No

Reviewer #3: Yes

4. Is the manuscript presented in an intelligible fashion and written in standard English?

Reviewer #1: Yes

Reviewer #2: No

Reviewer #3: No

5. Review Comments to the Author

Reviewer #1: 1.The study uses data from 2018 to 2021 but does not adequately justify the selection of this specific timeframe. It is unclear whether the period before 2018 might have provided a better baseline for comparison, considering the potential long-term trends in agricultural employment that were disrupted by the pandemic. This omission could lead to a misinterpretation of the pandemic's actual impact.

2. The study is centered on Brazilian municipalities, but the discussion does not sufficiently address whether or how the findings might be applicable to other contexts or countries. This limits the broader relevance of the research and its contribution to global discussions on agricultural employment during crises.

3. The paper's methodology section lacks a thorough discussion of potential biases in the data or limitations in the spatial models used. For example, the impact of missing data or inaccuracies in reported COVID-19 cases and deaths on the results is not considered, which could affect the validity of the findings.

4. The literature review is somewhat superficial and does not fully engage with the existing body of work on agricultural employment or the broader socio-economic impacts of pandemics.

5. The interpretation of the results, especially regarding the impact of COVID-19 on employment, is occasionally ambiguous. The paper sometimes conflates correlation with causation, suggesting that the pandemic directly caused certain employment trends without fully substantiating these claims.

Reviewer #2: Some Comments on PONE-D-24-24987

Dear author,

Congratulations on completing an excellent manuscript, which examines the effects of the COVID-19 pandemic on agricultural employment in Brazilian municipalities. The effort you put into the manuscript has paid off. I consider it an excellent manuscript, so I will try my best to provide some modification suggestions for your reference.

[1] The paper contains several typographical and grammatical errors, poor sentence structure, and overlooks simple tenets of standard writing. Please check.

[2] Let's start with the title 「The Effects of the COVID-19 Pandemic on Agricultural Employment in Brazilian Municipalities」. In my opinion, the title of this manuscript is not attractive enough because it only reflects the existence of some influencing factors, but the expression of specific factors is not clear enough. In other words, the COVID-19 Pandemic is not only a simple factor. It contains a string of elements, which you have also pointed out in the following text.

[3] The paper constantly makes use of improper pronouns such as "we" which should be avoided in scientific studies.

[4] The marginal contribution of this manuscript is not clearly stated. If necessary, it is recommended to separately state the marginal contribution of the manuscript in the literature review or introduction section. Is this manuscript just applying other research methods to the data in Brazilian Municipalities? If it's convenient, please reply to me with your thoughts.

[5] The references cited in this manuscript lack authoritative journal papers (e.g. Papers from American Economic Review(AER), Quarterly Journal of Economics(QJE), Journal of Political Economy(JPE)). Some old references in the literature should be replaced with most recent (past 5 years) studies. Please check it.

[6] The literature review section of this manuscript lacks some content. Specifically, the authors should conduct a more in-depth analysis of the reasons for changes in labor allocation in the production sector, rather than simply summarizing the conclusions of other literature.

[7] Starting from Line 215, the authors explain the calculation method of the index. But in my opinion, these are relatively basic econometric contents that are not original to the authors, and most scholars are familiar with them. In other words, they can be deleted.

[8] This manuscript involves empirical research, which involves hypothesis testing. I suggest the authors clearly state before Line 200 which hypotheses are proposed in this manuscript, which will also facilitate readers' reading. You can just use “H1: xxx; H2: xxx” to illustrate it.

[9] My question goes as: Is your suggestion universally applicable? Or just targeting the Brazilian Municipalities? Plus, the outlook for future research is not enough. It is needed to supplement and improve this section.

[10] We all know that the epidemic began in 2019. The data you are using starts from 2018. In my opinion, the starting year of this data should be earlier than 2018, and the ending year should be later than 2021 (compared to August 2024 when I received this manuscript).

[11] Please provide an explanation of the sample size.

[12] Figure 1 in the manuscript clearly illustrates the relevant content. I suggest adding a graph to illustrate the changes (often called △/ delta) rather than absolute values.

[13] In Table 5, you have mentioned the “x2”. Do you mean “χ2”? It is called “Chi-square”.

Once again, congratulations on completing such a high-quality paper. Wishing your manuscript a speedy publication on PLoS One.

Reviewer #3: Reviewer Comments

This paper investigates the impact of the COVID-19 pandemic on agricultural employment in Brazil, employing spatial econometric models to analyze the relevant factors. The research holds both theoretical and practical significance, but there are several areas for improvement.

Specific Comments

1. Introduction:

• The research question needs to be further clarified. For example, what is the extent of the impact of the COVID-19 pandemic on agricultural in Brazil（Not just employment rates, because you want to highlight the value of your research）? Which factors have the most significant impact on agricultural employment?

• Consider adding an introduction to the current status of agricultural development in Brazil to better understand the research background.

2. Literature Review:

• It is recommended to supplement some international comparative studies on the impact of the COVID-19 pandemic on agricultural employment and research on the application of spatial econometric models in agricultural employment studies. Compare the differences between their studies and the advantages of ours.

• Consider categorizing the literature review by research themes, such as the impact of the COVID-19 pandemic on agricultural employment, factors influencing agricultural employment, and the application of spatial econometric models.

3. Research Methodology:

• It is recommended to detail the construction method and selection rationale for the spatial weight matrix, as well as the specific steps and results of model selection.

4. Research Data:

• The research data is comprehensive but still needs further improvement. It is recommended to explain the data collection method, the reliability of the data source, and the handling method of missing data.

o Consider adding some methods for evaluating data quality, such as descriptive statistical analysis and data cleaning.

5. Model Selection:

• It is recommended to explain the reasons for choosing the Spatial Durbin Model and the robustness test of the model selection results.

• Consider adding an introduction to other model selection methods, such as cross-validation and information criteria.

6. Research Results:

• It is recommended to explain the economic interpretation of the research results and the implications for theory and practice.

• Consider adding some robustness tests of the research results, such as sensitivity analysis and heterogeneity analysis.

7. Heterogeneity Analysis:

• It is recommended to add heterogeneity analysis, such as group analysis by region, agricultural type, and labor type, to reveal whether the impact of the COVID-19 pandemic on agricultural employment in Brazil varies.

8. Conclusion:

• The conclusion is concise but still needs further improvement. It is recommended to summarize the main findings of the research and point out the future research directions.

Additional Suggestions

• It is recommended that the authors carefully revise the paper format to ensure compliance with the journal requirements.

• It is recommended that the authors strengthen the English expression of the paper to improve readability.

This paper has research value but still needs improvement. It is hoped that the authors can revise the paper according to the comments of the reviewers to improve the quality of the paper.

6. PLOS authors have the option to publish the peer review history of their article (what does this mean?). If published, this will include your full peer review and any attached files.

Reviewer #1: No

Reviewer #2: **Yes: **Qiaoyu Chen

Reviewer #3: **Yes: **Xiansheng Chen

---

## [Author Response · Author response to Decision Letter 0]

27 Nov 2024

Dear Dr. Carolina Serpieri and Reviewers,

We are grateful for the time and valuable contributions that each of you made to review our manuscript. Your suggestions were essential to identifying areas for improvement and increasing the robustness and clarity of our research. In response, we revised the manuscript in detail. We believe that incorporating your recommendations helped to strengthen the focus, broaden the scope of the literature review, and improve methodological accuracy, contributing to the results making a significant contribution to the topic at hand. Below is a summary of the changes made in response to the suggestions received.

Response to Editor's Comments:

Suggestion 1: “Introduction:

The research question needs to be further clarified. For example , what is the extent of the impact of the COVID-19 pandemic on agricultural in Brazil （ Not just employment rates, because you want to highlight the value of your research ） ? Which factors have the most significant impact on agricultural employment ?

Consider adding an introduction to the current status of agricultural development in Brazil to better understand the research background.”

Answer: We have reworded the introduction to expand the focus of the research proposal beyond the impact of the pandemic. We have also included general data that highlight the importance of the agricultural sector in the Brazilian economy, such as its contribution to the Gross Domestic Product (GDP) and formal employment. We have made the following points as suggested: we have incorporated information on the role of the agricultural sector in the Brazilian economy, we have specified more clearly the objective of the study, detailing the determinants analyzed in the variation of agricultural employment, we have justified the use of a multifactorial approach to understand the various influences on the sector. In addition, we have highlighted the importance of spatial analysis to capture regional interactions and variations in the distribution of agricultural employment.

Suggestion 2: “ Literature Review : It is recommended to supplement some international comparative studies on the impact of the COVID-19 pandemic on agricultural employment and research on the application of spatial econometric models in agricultural employment studies. Compare the differences between their studies and the advantages of ours .

Consider categorizing the literature review by research themes , such as the impact of the COVID-19 pandemic on agricultural employment , factors influencing agricultural employment , and the application of spatial econometric models.”

Answer: In the theoretical framework, we made a significant restructuring, organizing the literature review according to the categorization of the topics covered, as suggested. We added a specific section to discuss the results of international studies that applied spatial methods similar to ours, allowing for a comparative analysis. We also expanded the references, including more recent and relevant studies published in prominent journals. In addition, we incorporated figures and maps to enrich the content with visual elements, complementing the textual analysis.

Suggestion 3 – “ Research Data: The research data is comprehensive but still needs further improvement . It is recommended to explain the data collection method , the reliability of the data source , and the handling method of missing data.

Consider adding some methods for evaluating data quality , such as descriptive statistical analysis and data cleaning .”

Response: The contributions have been incorporated. We have added two justifications for using the spatial weights matrix and completely reworked the section on spatial model selection, using Elhorst 's (2010) approach to select the model that best fits the data.

Suggestions related to method and results. Model Selection ; Research Results : Heterogeneity Analysis : Conclusion ;

All suggestions for methodological improvements and presentation of results have been addressed. We have explained in more detail the reliability of the data sources and the treatment given to missing data. We have also added a new section in the methodology to describe the heterogeneity and robustness tests applied, with the results presented afterwards. The conclusions have been improved to reflect these additions and strengthen the final analysis.

Response to Reviewer 1's Comments:

Suggestion 1: “The study uses data from 2018 to 2021 but does not adequately justify the selection of this specific timeframe . It is unclear whether the period before 2018 might have provided a better baseline for comparison , considering the potential long-term trends in agriculture employment that were disrupted by the pandemic . This omission could lead to a misinterpretation of the pandemic's current impact . ”

Answer: We have supplemented the specific section of the methodology “ Data description” by reinforcing the justification for choosing the time frame. We emphasize that the inclusion of periods prior to 2018 could distort the analysis of the results, given the significant effects of the crisis that hit Brazil between 2014 and 2016. 

The selection of the period from 2018 to 2021 was strategically chosen with the aim of capturing both the pre-existing effects and the direct impact of the COVID-19 pandemic on the agricultural sector and on the municipal assistance policies implemented during the pandemic. The choice of this interval allows for a consistent analysis of the economic and social conditions immediately prior to the pandemic, establishing an appropriate baseline for comparison. From an analytical point of view, introducing years prior to 2018 could result in the generation of significant biases, since events such as the 2015-2016 economic crisis also had a profound impact on the agricultural sector and rural employment in Brazil, and such an effect could make it difficult to accurately distinguish the specific implications of the pandemic. Furthermore, the period covered by the work is in line with the availability of official data when the work was in the database construction phase.

Suggestion 2: “ The study is centered on Brazilian municipalities , but the discussion does not sufficiently address whether or how the findings might be applicable to other contexts or countries. This limits the broader relevance of the research and its contribution to global discussions on agricultural employment during crises. ”

Response: We have included a discussion of the generality of the results in the conclusion section, noting that although the focus was on Brazilian municipalities, the findings may have relevance for other developing countries with significant agricultural economies and similar social structures. Contexts in which the agricultural sector plays a key role may observe similar patterns of resilience, especially in areas where seasonal employment and exports are relevant.

However, it is important to note that the particularities of each country—such as the structure of production chains, specific public policies, and the degree of agricultural mechanization—influence how these factors interact with local employment. Thus, although the results of this research contribute to the debate on the impact of COVID-19 on the agricultural sector, their applicability to other contexts requires a careful examination of the economic and social specificities of each country.

Suggestion 3: “ The paper's methodology section lacks thoroughness discussion of potential biases in the data or limitations in the spatial models used . For example , the impact of missing data or inaccuracies in reported COVID-19 cases and deaths on the results is not considered , which could affect the validity of the findings . ”.

Answer: We have restructured the research variables section, highlighting the credibility of the official data sources and the technique applied to treat missing data (“ Data description”) . We opted for simple median imputation, given the low proportion of missing values (less than 3% per variable). This technique is less sensitive to outliers and preserves statistical power, avoiding the losses that would occur with the complete exclusion of cases, which maintains the integrity of the econometric models.

Suggestion 4: “ The literature review is somewhat superficial and does not fully engage with the existing body of work on agricultural employment or the broader socio-economic impacts of pandemics ”

Answer: We have completely reformulated the theoretical framework, highlighting the effects of the pandemic on agriculture, a section that addresses studies that used similar methods, the determinants of agricultural employment and a regional profile of agriculture in Brazil. In addition, we have included more current and relevant references.

Suggestion 5: “ The interpretation of the results , especially regarding the impact of COVID-19 on employment , is occasionally ambiguous . The paper sometimes conflates correlation with causation , suggesting that the pandemic directly caused certain employment trends without fully substantiating these claims . ”

Answer: We have comprehensively reformulated the interpretation of the results, making it clearer and unambiguous. In addition, we have further analyzed both the direct and indirect impacts for each of the variables included in the model.

Response to Reviewer 2's Comments:

Suggestion 1: “ The paper contains several typographical and grammatical errors, poor sentence structure, and overlooks simple tenets of standard writing. Please check .. ”

Answer: We carried out an extensive review of the English language, ensuring the quality of the text.

Suggestion 2: “ Let's start with the title “The Effects of the COVID-19 Pandemic on Agricultural Employment in Brazil Municipalities” . In my opinion , the title of this manuscript is not attractive enough because it only reflects the existence of some influencing factors , but the expression of specific factors is not clear enough . In other words, the COVID-19 Pandemic is not just a simple factor . It contains a string of elements , which you have also pointed out in the following text ”

Answer: We have reworded the title to: “ Determinants of Agricultural Employment During the COVID-19 Pandemic : A Spatial Analysis of Brazilian Municipalities ”. The idea is to highlight that we investigated several dimensions that influence agricultural employment, in addition to the covid-19 pandemic, as well as to show that this is an analysis that considers spatial aspects.

Suggestion 3: “ The paper constantly makes use of improper pronouns such as " we " which should be avoided in scientific studies . ”

Answer: We reviewed the text and removed inappropriate pronouns from the text.

Suggestion 4: “ The marginal contribution of this manuscript is not clearly stated . If necessary , it is recommended to separately state the marginal contribution of the manuscript in the literature review or introduction section . Is this manuscript just applying other research methods to the data in Brazilian Municipalities ? If it's convenient , please reply to me with your thoughts . ”.

Answer: In order to address the marginal contribution of the manuscript, we have made changes to the introduction to better highlight the main points. In the third paragraph, we specify that the contribution of the study is threefold. First, we investigate the socioeconomic determinants, public policies, extreme weather events, and the impact of COVID-19 on agricultural employment, something that has not been explored in the literature with a spatial approach and focused on Brazil. Second, we incorporate a spillover analysis at the municipal level, allowing us to observe the direct and indirect impacts of these factors on the spatial distribution of agricultural employment, something essential to understand local economic interactions. Finally, the analysis allows for a regional differentiation that highlights how economic characteristics and weather events heterogeneously affect agricultural employment across municipalities.

In the theoretical framework, we include a section that highlights the gap in spatial studies focused on the effects of the pandemic on agricultural employment in Brazil, a significant marginal contribution of our study.

Suggestion 5: “ The references cited in this manuscript lack authoritative journal papers (eg Papers from American Economic Review ( AER), Quarterly Journal of Economics (QJE), Journal of Political Economy (JPE)). Some old references in the literature should be replaced with most recent ( past 5 years ) studies . Please check it ”.

Answer : We inserted three articles from the suggested journals into the theoretical framework, namely: Zabatantou et al. (2023) , Yeboah et al. (2021) and Fahokhi & Pellegrina (2023) . In addition to these, we added numerous other references to more recent articles published in relevant journals.

Suggestion 6: “ The literature review section of this manuscript lacks some content . Specifically , the authors should conduct a more in-depth analysis of the reasons for changes in labor allocation in the production sector, rather than simply summarizing the conclusions of other literature .”

Answer: We have reworded the section that presents the literature review, including highlighting the issue of the migration of workers from industry and services to agriculture. To this end, we used as references the works of Deb (2021) , focusing on India, and Charlton & Castillo (2020) focusing on the United States of America (USA). With regard to Brazil, we have not found any relevant studies that investigate this phenomenon to date.

Suggestion 7 : “ Starting from Line 215, the authors explain the calculation method of the index. But in my opinion , these are relatively basic econometric contents that are not original to the authors , and most scholars are familiar with them . In other words, they can be deleted . ”

Answer: We have accepted the suggestion and simplified the section on the spatial autocorrelation index, making it more concise and straightforward, as can be seen in the section " Spatial Correlation Test: Moran's Index".

Suggestion 8: “ This manuscript involves empirical research , which involves hypothesis testing . I suggest the authors clearly state before Line 200 which hypotheses are proposed in this manuscript , which will also facilitate readers ' reading . You can just use “H1: xxx ; H2: xxx ” to illustrate it ”

Answer: We accept the suggestion and insert 5 research hypotheses in the sections of the theoretical framework. They are:

“H 1 : The pandemic , in terms of the number of COVID-19 cases, positively impacted agricultural employment in Brazil . ( Impact Section of the COVID-19 Pandemic on Agricultural Employment ).

H 2 : The effects of the COVID-19 pandemic on agricultural employment present spatial spillover effects , both at the macro and regional levels . ( Application section of Spatial Econometric Models)

H 3 and H 4 : Extreme weather events had a significant impact on the reduction of agricultural employment during the period and H4: The implementation of public income transfer policies during the health crisis positively influenced the maintenance of agricultural employment . ( Determinants section of Agricultural Employment ).

H 5 : The effects of the pandemic on agricultural employment is heterogeneous , varying according to the productive characteristics of each region of the country (Regional Characteristics Section) of the Agricultural Sector in Brazil ).

In addition to the hypotheses, a figure was included (“Figure 2”) to highlight and improve visual understanding of these points.

Suggestion 9: “ My question goes as: Is your suggestion universally applicable? Or just targeting the Brazilian Municipalities? Plus, the outlook for future research is not enough. It is needed to supplement and improve this section .. ”

Answer: We adopted the proposed suggestions, and such considerations are set out in the conclusion of the study. Although our study focuses on Brazilian municipalities, the results may be applicable to other developing countries with similar agricultural economies, especially those in 

---

## [Editor Report · Decision Letter 1]

5 Dec 2024

PONE-D-24-24987R1Determinants of Agricultural Employment During the COVID-19 Pandemic: a Spatial Analysis of Brazilian MunicipalitiesPLOS ONE

Dear Dr. Alvim,

Thank you for submitting your manuscript to PLOS ONE. After careful consideration, we feel that it has merit but does not fully meet PLOS ONE’s publication criteria as it currently stands. Therefore, we invite you to submit a revised version of the manuscript that addresses the points raised during the review process.

Specifically, the tables should be better formatted to ensure consistency and uniformity throughout the manuscript. Using a homogeneous style for all tables will enhance the manuscript’s readability.

Additionally, we recommend providing high-resolution versions of all figures to ensure they are clear and of publication quality. This will greatly improve the visual presentation of your work and its accessibility to readers.

We look forward to receiving these minor revisions and remain confident that they will significantly enhance the overall quality of your manuscript. 

We look forward to receiving your revised manuscript.

Kind regards,

Carolina Serpieri, PhD

Academic Editor

PLOS ONE
---

## [Author Response · Author response to Decision Letter 1]

10 Dec 2024

Response to the editor: Second round of revisions.

Suggestion 1: "The tables should be better formatted to ensure consistency and uniformity throughout the manuscript. Using a homogeneous style for all tables will enhance the readability of the manuscript."

Response: We made adjustments to the last two tables in the article, which were added during the revision process, to ensure they align with the style of the other tables and meet the journal's formatting specifications. As a result, we have ensured consistency and uniformity in the table formats, aiming to improve the manuscript's readability.

Suggestion 2: "We recommend providing high-resolution versions of all figures to ensure they are clear and of publication quality. This will greatly improve the visual presentation of your work and its accessibility to readers."

Response: We adjusted the figures according to the journal's specifications and used the "Preflight Analysis and Conversion Engine (PACE)" tool to ensure all images meet the required quality standards. This ensures that the figures are now in high resolution, providing an enhanced visual presentation and greater clarity for the readers.

Additionally, we conducted a detailed review of the references and identified three studies cited in the main text but not included in the reference list: Leivas et al. (2015), Conab (2020), and Li et al. (2021). We added the first two studies to the reference list and replaced the reference to Li et al. (2021) with Bivand et al. (2021), as we believe this substitution enhances the value of the research due to the greater relevance of Bivand's study.

Response to Editor's Comments (First round):

Suggestion 1: “Introduction:

The research question needs to be further clarified. For example , what is the extent of the impact of the COVID-19 pandemic on agricultural in Brazil （ Not just employment rates, because you want to highlight the value of your research ） ? Which factors have the most significant impact on agricultural employment ?

Consider adding an introduction to the current status of agricultural development in Brazil to better understand the research background.”

Answer: We have reworded the introduction to expand the focus of the research proposal beyond the impact of the pandemic. We have also included general data that highlight the importance of the agricultural sector in the Brazilian economy, such as its contribution to the Gross Domestic Product (GDP) and formal employment. We have made the following points as suggested: we have incorporated information on the role of the agricultural sector in the Brazilian economy, we have specified more clearly the objective of the study, detailing the determinants analyzed in the variation of agricultural employment, we have justified the use of a multifactorial approach to understand the various influences on the sector. In addition, we have highlighted the importance of spatial analysis to capture regional interactions and variations in the distribution of agricultural employment.

Suggestion 2: “ Literature Review : It is recommended to supplement some international comparative studies on the impact of the COVID-19 pandemic on agricultural employment and research on the application of spatial econometric models in agricultural employment studies. Compare the differences between their studies and the advantages of ours .

Consider categorizing the literature review by research themes , such as the impact of the COVID-19 pandemic on agricultural employment , factors influencing agricultural employment , and the application of spatial econometric models.”

Answer: In the theoretical framework, we made a significant restructuring, organizing the literature review according to the categorization of the topics covered, as suggested. We added a specific section to discuss the results of international studies that applied spatial methods similar to ours, allowing for a comparative analysis. We also expanded the references, including more recent and relevant studies published in prominent journals. In addition, we incorporated figures and maps to enrich the content with visual elements, complementing the textual analysis.

Suggestion 3 – “ Research Data: The research data is comprehensive but still needs further improvement . It is recommended to explain the data collection method , the reliability of the data source , and the handling method of missing data.

Consider adding some methods for evaluating data quality , such as descriptive statistical analysis and data cleaning .”

Response: The contributions have been incorporated. We have added two justifications for using the spatial weights matrix and completely reworked the section on spatial model selection, using Elhorst 's (2010) approach to select the model that best fits the data.

Suggestions related to method and results. Model Selection ; Research Results : Heterogeneity Analysis : Conclusion ;

All suggestions for methodological improvements and presentation of results have been addressed. We have explained in more detail the reliability of the data sources and the treatment given to missing data. We have also added a new section in the methodology to describe the heterogeneity and robustness tests applied, with the results presented afterwards. The conclusions have been improved to reflect these additions and strengthen the final analysis.

Response to Reviewer 1's Comments:

Suggestion 1: “The study uses data from 2018 to 2021 but does not adequately justify the selection of this specific timeframe . It is unclear whether the period before 2018 might have provided a better baseline for comparison , considering the potential long-term trends in agriculture employment that were disrupted by the pandemic . This omission could lead to a misinterpretation of the pandemic's current impact . ”

Answer: We have supplemented the specific section of the methodology “ Data description” by reinforcing the justification for choosing the time frame. We emphasize that the inclusion of periods prior to 2018 could distort the analysis of the results, given the significant effects of the crisis that hit Brazil between 2014 and 2016. 

The selection of the period from 2018 to 2021 was strategically chosen with the aim of capturing both the pre-existing effects and the direct impact of the COVID-19 pandemic on the agricultural sector and on the municipal assistance policies implemented during the pandemic. The choice of this interval allows for a consistent analysis of the economic and social conditions immediately prior to the pandemic, establishing an appropriate baseline for comparison. From an analytical point of view, introducing years prior to 2018 could result in the generation of significant biases, since events such as the 2015-2016 economic crisis also had a profound impact on the agricultural sector and rural employment in Brazil, and such an effect could make it difficult to accurately distinguish the specific implications of the pandemic. Furthermore, the period covered by the work is in line with the availability of official data when the work was in the database construction phase.

Suggestion 2: “ The study is centered on Brazilian municipalities , but the discussion does not sufficiently address whether or how the findings might be applicable to other contexts or countries. This limits the broader relevance of the research and its contribution to global discussions on agricultural employment during crises. ”

Response: We have included a discussion of the generality of the results in the conclusion section, noting that although the focus was on Brazilian municipalities, the findings may have relevance for other developing countries with significant agricultural economies and similar social structures. Contexts in which the agricultural sector plays a key role may observe similar patterns of resilience, especially in areas where seasonal employment and exports are relevant.

However, it is important to note that the particularities of each country—such as the structure of production chains, specific public policies, and the degree of agricultural mechanization—influence how these factors interact with local employment. Thus, although the results of this research contribute to the debate on the impact of COVID-19 on the agricultural sector, their applicability to other contexts requires a careful examination of the economic and social specificities of each country.

Suggestion 3: “ The paper's methodology section lacks thoroughness discussion of potential biases in the data or limitations in the spatial models used . For example , the impact of missing data or inaccuracies in reported COVID-19 cases and deaths on the results is not considered , which could affect the validity of the findings . ”.

Answer: We have restructured the research variables section, highlighting the credibility of the official data sources and the technique applied to treat missing data (“ Data description”) . We opted for simple median imputation, given the low proportion of missing values (less than 3% per variable). This technique is less sensitive to outliers and preserves statistical power, avoiding the losses that would occur with the complete exclusion of cases, which maintains the integrity of the econometric models.

Suggestion 4: “ The literature review is somewhat superficial and does not fully engage with the existing body of work on agricultural employment or the broader socio-economic impacts of pandemics ”

Answer: We have completely reformulated the theoretical framework, highlighting the effects of the pandemic on agriculture, a section that addresses studies that used similar methods, the determinants of agricultural employment and a regional profile of agriculture in Brazil. In addition, we have included more current and relevant references.

Suggestion 5: “ The interpretation of the results , especially regarding the impact of COVID-19 on employment , is occasionally ambiguous . The paper sometimes conflates correlation with causation , suggesting that the pandemic directly caused certain employment trends without fully substantiating these claims . ”

Answer: We have comprehensively reformulated the interpretation of the results, making it clearer and unambiguous. In addition, we have further analyzed both the direct and indirect impacts for each of the variables included in the model.

Response to Reviewer 2's Comments:

Suggestion 1: “ The paper contains several typographical and grammatical errors, poor sentence structure, and overlooks simple tenets of standard writing. Please check .. ”

Answer: We carried out an extensive review of the English language, ensuring the quality of the text.

Suggestion 2: “ Let's start with the title “The Effects of the COVID-19 Pandemic on Agricultural Employment in Brazil Municipalities” . In my opinion , the title of this manuscript is not attractive enough because it only reflects the existence of some influencing factors , but the expression of specific factors is not clear enough . In other words, the COVID-19 Pandemic is not just a simple factor . It contains a string of elements , which you have also pointed out in the following text ”

Answer: We have reworded the title to: “ Determinants of Agricultural Employment During the COVID-19 Pandemic : A Spatial Analysis of Brazilian Municipalities ”. The idea is to highlight that we investigated several dimensions that influence agricultural employment, in addition to the covid-19 pandemic, as well as to show that this is an analysis that considers spatial aspects.

Suggestion 3: “ The paper constantly makes use of improper pronouns such as " we " which should be avoided in scientific studies . ”

Answer: We reviewed the text and removed inappropriate pronouns from the text.

Suggestion 4: “ The marginal contribution of this manuscript is not clearly stated . If necessary , it is recommended to separately state the marginal contribution of the manuscript in the literature review or introduction section . Is this manuscript just applying other research methods to the data in Brazilian Municipalities ? If it's convenient , please reply to me with your thoughts . ”.

Answer: In order to address the marginal contribution of the manuscript, we have made changes to the introduction to better highlight the main points. In the third paragraph, we specify that the contribution of the study is threefold. First, we investigate the socioeconomic determinants, public policies, extreme weather events, and the impact of COVID-19 on agricultural employment, something that has not been explored in the literature with a spatial approach and focused on Brazil. Second, we incorporate a spillover analysis at the municipal level, allowing us to observe the direct and indirect impacts of these factors on the spatial distribution of agricultural employment, something essential to understand local economic interactions. Finally, the analysis allows for a regional differentiation that highlights how economic characteristics and weather events heterogeneously affect agricultural employment across municipalities.

In the theoretical framework, we include a section that highlights the gap in spatial studies focused on the effects of the pandemic on agricultural employment in Brazil, a significant marginal contribution of our study.

Suggestion 5: “ The references cited in this manuscript lack authoritative journal papers (eg Papers from American Economic Review ( AER), Quarterly Journal of Economics (QJE), Journal of Political Economy (JPE)). Some old references in the literature should be replaced with most recent ( past 5 years ) studies . Please check it ”.

Answer : We inserted three articles from the suggested journals into the theoretical framework, namely: Zabatantou et al. (2023) , Yeboah et al. (2021) and Fahokhi & Pellegrina (2023) . In addition to these, we added numerous other references to more recent articles published in relevant journals.

Suggestion 6: “ The literature review section of this manuscript lacks some content . Specifically , the authors should conduct a more in-depth analysis of the reasons for changes in labor allocation in the production sector, rather than simply summarizing the conclusions of other literature .”

Answer: We have reworded the section that presents the literature review, including highlighting the issue of the migration of workers from industry and services to agriculture. To this end, we used as references the works of Deb (2021) , focusing on India, and Charlton & Castillo (2020) focusing on the United States of America (USA). With regard to Brazil, we have not found any relevant studies that investigate this phenomenon to date.

Suggestion 7 : “ Starting from Line 215, the authors explain the calculation method of the index. But in my opinion , these are relatively basic econometric contents that are not original to the authors , and most scholars are familiar with them . In other words, they can be deleted . ”

Answer: We have accepted the suggestion and simplified the section on the spatial autocorrelation index, making it more concise and straightforward, as can be seen in the section " Spatial Correlation Test: Moran's Index".

Suggestion 8: “ This manuscript involves empirical research , which involves hypothesis testing . I suggest the authors clearly state before Line 200 which hypotheses are proposed in this manuscript , which will also facilitate readers ' reading . You can just use “H1: xxx ; H2: xxx ” to illustrate it ”

Answer: We accept the suggestion and insert 5 research hypotheses in the sections of the theoretical framework. They are:

“H 1 : The pandemic , in terms of the number of COVID-19 cases, positively impacted agricultural employment in Brazil . ( Impact Section of the COVID-19 Pandemic on Agricultural Employment ).

H 2 : The effects of the COVID-19 pandemic on agricultural employment present spatial spillover effects , both at the macro and regional levels . ( Application section of Spatial Econometric Models)

H 3 and H 4 : Extreme weather events had a significant impact on the reduction of agricultural employment during the period and H4: The implementation of public income transfer policies during the health crisis pos

---

## [Editor Report · Decision Letter 2]

11 Dec 2024

Determinants of Agricultural Employment During the COVID-19 Pandemic: a Spatial Analysis of Brazilian Municipalities

PONE-D-24-24987R2

Dear Dr. Augusto Mussi Alvim,

We’re pleased to inform you that your manuscript has been judged scientifically suitable for publication and will be formally accepted for publication once it meets all outstanding technical requirements.

Kind regards,

Carolina Serpieri, PhD

Academic Editor

PLOS ONE
---

## [Editor Report · Acceptance letter]

27 Dec 2024

PONE-D-24-24987R2 

PLOS ONE

Dear Dr. Alvim, 

I'm pleased to inform you that your manuscript has been deemed suitable for publication in PLOS ONE. Congratulations! Your manuscript is now being handed over to our production team.

Kind regards, 

on behalf of

Dr. Carolina Serpieri 

Academic Editor

PLOS ONE